# Mining LDA topics on construction engineering change risks based on graded evidence

Lianghai Jin[1,2,3,4]*, Chenxi Li[1,2], Zhongrong Zhu[1,2,4], Songxiang Zou[1,2], Xushu Sun[1,2]

**1** Hubei Key Laboratory of Hydro-Power Construction and Management, China Three Gorges University, Yichang, Hubei, China, **2** College of Hydraulic and Environmental Engineering, China Three Gorges University, Yichang, Hubei, China, **3** Construction Law Appraisal Center, China Three Gorges University, Yichang, Hubei, China, **4** Safety Production Standardization Review Center, China Three Gorges University, Yichang, Hubei, China

* jinlianghai@ctgu.edu.cn

**Data Availability Statement:** All documents are available on the China Judicial Documents Network database (https://wenshu.court.gov.cn/website/wenshu/181010CARHS5BS3C/index.html?open=login).

## Abstract

Engineering change (EC) risk may negatively impact project schedule, cost, quality, and stakeholder satisfaction. However, existing methods for managing EC risk have certain shortcomings in evidence selection and do not adequately consider the quality and reliability of evidence associated with EC risks. Evidence grading plays a crucial role in ensuring the reliability of decisions related to EC risks and can provide essential scientific and reliability support for decision-making. In order to explore the potential risks associated with architectural engineering changes (ECs) and identify the most significant ones, this study proposed a methodology that combines evidence grading theory and Latent Dirichlet Allocation (LDA) topic analysis means. Initially, the evidence-based grading theory served as the creation of a grading table for evidence sources related to EC risk. Specifically, we categorized the evidence sources into three levels based on their credibility. Subsequently, we selected evidence with higher credibility levels for textual analysis, utilizing the LDA topic model. This involved analyzing regulations, industry standards, and judgment documents related to EC, ultimately identifying the themes associated with EC risks. In addition, by combining EC risk topics with relevant literature, we identified factors influencing EC risks. Subsequently, we designed an expert survey questionnaire to determine the key risks and important risk topics associated with potential risks. The results show that by synthesizing information from both Class A and B evidence, a total of five prominent risk themes were identified, namely contract, technology, funds, personnel, and other hazards. Among them, the technical risk has the highest value, so it implies that the risk is the most important, and the key risks are engineering design defects, errors, and omissions.

## 1. Introduction

In the field of construction project management, EC is an important topic that not only has a wide-ranging impact on the progress of a project but also poses a major challenge to its

**Funding:** Department of Humanities and Social Affairs Fund of the Ministry of Education (21YJA630038). Their contributions during the research design phase and active involvement in manuscript preparation are greatly appreciated.

**Competing interests:** The authors have declared that no competing interests exist.

successful implementation. This is mainly due to the fact that construction projects are usually accompanied by high investment, a long construction period, and complexity affected by a variety of natural conditions and objective factors. The frequent occurrence of ECs has become the norm in the industry [1, 2], and these changes may trigger a series of undesirable consequences, such as delays in project duration, uncontrolled investment, difficulty in controlling quality, and an increase in safety hazards [3]. In addition, project changes are also prone to problems such as contract disputes, construction claims, and litigation [4, 5]. The risk identification of ECs requires potential evidence-based grading, whereas evidence-based grading is derived from evidence-based management [6, 7].

Evidence-based management is a scientific development mechanism for summarizing the past, discovering new problems, conducting scientific research, and creating evidence to solve problems [14]. Similarly, EC risk management is based on summarizing experience combined with engineering reality, discovering risk factors, scientifically analyzing risks, identifying key risks, and implementing risk assessment and control. The main idea behind evidence is that people who make decisions should use scientifically sound evidence that comes from real-world research and testing, and that they should fully and effectively use this evidence. At the same time, the main goal is to combine original evidence to get better evidence by deeply examining the parts of evidence from many different sources [8]. Obtaining high-quality evidence is not only a key element in the proof-building process but also an important guarantee for ensuring the reliability of results. Therefore, there is a need for the grading of evidence.

The majority of data pertaining to risks associated with ECs predominantly exists in textual form. Text mining methods possess a strong capability in extracting and analyzing vast amounts of unstructured text data [9]. Specifically, the LDA thematic model has demonstrated significant efficacy in text data modeling [10]. As a result, this research proposes an LDA text analysis method based on evidence grading that focuses on mining risk themes relevant to ECs. Our research aims to thoroughly consider the quality and reliability of evidence associated with EC risks, explore potential risks related to Architectural ECs, and identify the most significant factors among them.

Our main contributions can be summarized as follows:

1. This study integrates evidence grading and text mining methodologies to intelligently extract textual data related to EC risk events. This integration enhances the quality and reliability of evidence in risk analysis, thereby promoting data-driven decision-making in risk management.

2. This study introduces the theory of evidence classification to the study of engineering change risk, focusing on the study of high-confidence evidence, which improves the scientific and reliability of the study.

3. Compared to Bayesian networks, the proposed method is more efficient and straightforward in handling large-scale textual data. By simplifying the sampling process, such as using Gibbs sampling, the analysis efficiency has been enhanced.

The remainder of this study is organized as follows: Section 2 primarily focuses on the relevant literature of this paper. Proposed methods and models are illustrated in Section 3. In Section 4, we delve into the model results. Evaluations results are discussed in Section 5, and Section 6 concludes this study.

## 2. Related works

### 2.1 Evidence grading theory

Nowadays, many scholars at home and abroad apply evidence-grading to scientific research. The international GRADE Working Group, based on examples, divided the quality of evidence into four levels: high, medium, low, and extremely low, in multiple dimensions, to assess the strength and reliability of the evidence [11]. Different research methodologies and a combination of evidence from multiple sources were emphasized to ensure that the conclusions were scientific and rigorous [12]. In general, the composition of evidence includes not only the digitization and encoding of facts but also expert opinions, research reports, research findings, statistical findings, etc [13]. Afzal created a quality-based, contextually perceived gradient of evidence to improve the quality of evidence in decision-making [14]. In China, Feng Jiahao built a pyramid of gradients of evidence sources, integrating evidence-based decision-making ideas into the DIIS process——a systematic approach encompassing the collection of Data, the revealing of Information, the synthesizing of Intelligence, and the formulation of a Solution and enhancing the efficiency and scientific of decision-making [15], and Zhang Haitao et al. [16] by organizing evidence levels derived from construction safety incidents, established an evidence database aimed at minimizing the risk of erroneous decisions caused by the randomness of experience and the tendency to ignore evidence. Rita Yi Man Li applies evidence-based practices to research on construction safety risks in Hong Kong, China. She ranks and compares renovation, maintenance, and repair projects with new skyscraper construction, considering the causes of construction accidents under different types of evidence [17]. The aforementioned studies substantiate that the graded management of evidence can culminate in decisions characterized by both efficacy and scientific rigor.

### 2.2 Risk management

Innovative research in construction safety is an important concern for every country [18]. Risk management is a crucial component in the field of construction safety, aiming to enhance the successful implementation and delivery of projects. At present, the research on EC risk management is mainly through the analysis of engineering cases, identifying change risks, and determining the change risks that affect the cost [19]. However, the risks of ECs are diversified, and most of them are statistically analyzed using a small number of common cases. Most research evidence is relatively small and cannot be summarized comprehensively or their evidence is largely subjective. Murat Cevikbas et al. [20] identified the risk factors of disruption claim management through literature review and utilized a fuzzy analytic hierarchy process to assess the relationships between these factors. This comprehensive guidance aims to assist construction practitioners in navigating the complexities of the field. Yeasin et al. [21] used a dynamic Bayesian network method to predict change propagation risk and EC duration; Eltaief et al. [22] focused on the risk assessment of EC to enhance the prediction capabilities of EC risk management. According to the above knowledge, traditional risk management involves the use of Bayesian networks and subjective analysis.

### 2.3 Text mining

LDA stands as a probabilistic and generative statistical model devised for the purpose of topic modeling. Originating from the collaborative efforts of David M. Blei, Andrew Y. Ng, and Michael I. Jordan in 2003 [23], LDA has found extensive application in natural language processing and text analysis, specifically aimed at unveiling latent topics embedded within a corpus of documents [24]. Over the years, the LDA model has become a cornerstone not only in

literature knowledge mining [25] but has also demonstrated prowess in diverse domains such as topic discovery, Sentiment analysis [26] and topic evolution [27]. Its effectiveness lies in deciphering the semantic essence concealed within voluminous texts, facilitating a nuanced understanding of latent thematic narratives.

The versatility of the LDA model extends beyond theoretical applications, with numerous practical implementations in literature knowledge mining [23]. Noteworthy achievements include its adeptness in uncovering latent topics, enabling a profound exploration of intricate thematic structures. For instance, Lee et al. [28] conducted a study leveraging LDA text mining technology to analyze user-generated text, providing valuable insights into the risks associated with electric scooters. Their findings contributed to the development of design improvements aimed at enhancing global safety for users. Another illustrative application comes from Liyun Zeng et al. [29], who utilized the LDA method to perform sentiment analysis, extract keywords, and identify themes in Instagram posts addressing topics related to construction health and safety. The study's outcomes yielded a comprehensive understanding of public perceptions within the construction industry, offering valuable insights for formulating preventative measures to enhance awareness and reduce the occurrence of accidents.

Given the literature discussed above, our approach unfolds in the following manner: Initially, we retrieve textual evidence related to ECs, which includes laws, regulations, and legal adjudication documents. Subsequently, we assign grades to the evidence based on its source. Following this, we employ the LDA thematic model to meticulously unearth risk themes embedded in a voluminous array of EC textual data. Lastly, through the deployment of survey questionnaires, we quantify the probability and degree of hazards associated with EC risks. This enables us to identify pivotal risks and provides a solid scientific basis for managing the inherent risks associated with ECs.

## 3. Materials and methods

In this section, a detailed exposition of the methods employed in this study is provided, aiming to comprehensively and clearly elucidate the research process of EC risk analysis. There has been an exploratory exploration into the integration of LDA and evidence-based decision-making, and the results indicate that this approach demonstrates a certain level of practicality and innovation [30]. The study is primarily grounded in evidence grading theory, coupled with text mining methods, to systematically extract critical information related to EC risks. In Fig 1, we present a method analysis flowchart that illustrates key steps and decision pathways adopted throughout the entire research process.

### 3.1 EC risk evidence grading framework

Classifying evidence according to its credibility is crucial for its scientific and pragmatic application in risk management, particularly in the sphere of EC. Inherently, evidence possesses

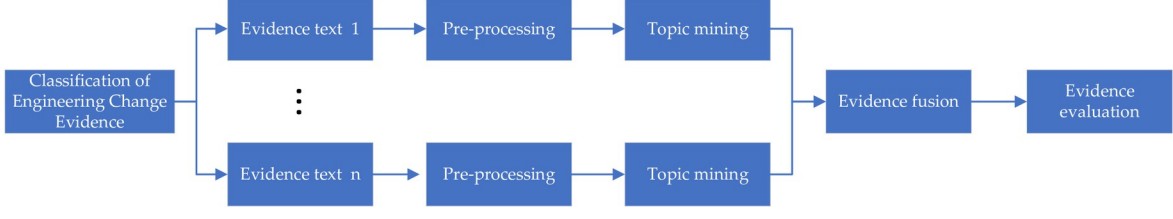

**Fig 1. EC risk analysis process.**

varying degrees of strength, thereby warranting categorization into distinct grades or levels. High-intensity evidence typically commands greater credibility and demonstrates pronounced guiding significance in practical scenarios [31]. Consequently, the stratification and grading of evidence serve to facilitate its more judicious and scientific use. In the pyramid system delineating evidence sources, the grading principle underscores a decrement in reliability and validity as evidence shifts from a high to a low grade; systematic evaluations, experimental studies, non-experimental research, case studies, and authoritative opinions progressively exhibit a reduced degree of recommendation [15]. Diverging from conventional evidence-based management science, EC risk evidence sources predominantly bifurcate into three categories: regulatory standards (class A), research literature (class B), and expert opinion (class C) [16]. Their respective persuasiveness is ranked as class A > class B > class C, further detailed in Table 1.

Considering the previously discussed classifications of evidence, the intrinsic confidence levels inherent to various sources unequivocally necessitate discerning application, particularly in the sphere of EC risk management. Evidence derived from varied sources inherently possesses disparate degrees of confidence, thereby influencing the reliability of inferential outcomes predicated on such evidence. It is pivotal to acknowledge that evidence boasting a higher confidence degree typically culminates in more robust conclusions. Consequently, applying evidence indiscriminately without a structured prioritization compromises the validity of the data. In the realm of EC risk identification, the strategic prioritization of evidence, especially that which holds a superior confidence level, becomes imperative. More precisely, evidence from class A regulatory standards and class B research literature ought to be accorded preferential consideration. Nonetheless, Type C evidence, which includes expert opinions, though potentially less convincing in some situations, still holds invaluable significance. The pertinence of expert opinion is underscored by its capacity to proffer invaluable insights and experiential sharing, gaining pronounced significance when predicated upon a substantial foundation of data and research findings. Furthermore, when the subjectivity of experts is supported by credible research outcomes, it plays a vital role in enabling evidence-based analysis of EC risks.

## 3.2 Data sources and processing

Analyzing the data concerning EC risk primarily involves dealing with text data, thereby necessitating the adoption of text analysis methodologies to unearth evidence pertaining to EC risk. Initially, the text data related to ECs undergoes a structured processing transition into high-quality information. Subsequently, through textual analysis, evidence is sifted and evaluated, eventually being transformed into valid proofs.

**Table 1. Classification and level of evidence sources in EC risk.**

| Evidence Conviction Level | Evidence of EC risks | |
|---|---|---|
| **High** | Class A: Regulatory standards | National laws. Administrative regulations, departmental rules, local regulations, local rules Industry standards and local standards. |
| **Medium** | Class B: Research literature | High-quality systematic review literature. Original data. Expert surveys, Delphi, and case studies. Descriptive studies such as cross-sectional and correlational studies. |
| **Low** | Class C: expert opinion | Opinions and consensus of expert panels and authoritative figures' opinions. |

**Table 2. EC related laws and regulations in China.**

| Number | EC related laws and regulations |
|---|---|
| 1 | *Management Measures For Design Change Of Water Conservancy Project* |
| 2 | *Interim Measures For Settlement Of Construction Project Price* |
| 3 | *Management Method Of Construction Project Quality Margin* |
| 4 | *The Bidding Law Of The People 'S Republic Of China* |
| 5 | *Construction Project Supervision Contract (Model Text)* |
| 6 | *Construction Law Of The People 'S Republic Of China* |
| 7 | *Civil Code Of The People 'S Republic Of China* |
| 8 | *Explanation Of The Supreme People 'S Court On The Application Of Legal Issues In The Trial Of Construction Contract Disputes* |
| 9 | *Interpretation Of The Supreme People 'S Court On The Application Of Legal Issues In The Trial Of Construction Contract Disputes* |

More specifically, after collecting data that reflects the EC process, evident sources are identified and hierarchically organized. This paper encompasses several channels of evident sources for EC, including regulatory standards, judicial documents, expert opinions, and more. Upon grading the evidence, a preference is established for the use of Category A evidence in the form of regulatory standard texts and Category B evidence found in judicial documents. After identifying and collecting the selected data, we identified nine legal regulatory texts as Category A evidence. EC related laws and regulations in China are presented in Table 2.

As for Category B evidence, we investigated EC judicial documents disclosed by China Judgement Online (website: https://wenshu.court.gov.cn/). To achieve effective filtering and processing, the following search criteria were established: " The search criteria included the information retrieval word '工程变更 (EC),' the case type 'civil case,' and the document type 'judgment.' The deadline is May 2023. " As a result, 265 EC judicial documents were initially retrieved. After a process of reading, screening, and removal of invalid texts, 193 valid judicial documents were compiled and summarized.

Text processing generally bifurcates into two segments. A customized word segmentation dictionary for EC text is created, which includes terms such as '工程变更 (EC),' '施工阶段 (construction phase),' and '工程造价 (construction cost).' This dictionary retains high-frequency words related to the theme for further analysis. The second phase focuses on data mining. The LDA topic model is used to extract document themes, while perplexity, in conjunction with pyLDAvis software, is employed to determine the optimal number of topics. This process facilitates the analysis of EC risk themes.

## 3.3 LDA topic analysis model

Topic modeling emerges as a crucial statistical analysis model in the realm of unsupervised machine learning, fundamentally serving document modeling [30]. Significantly, it has evolved into a primary paradigm in contemporary research pertaining to textual representation. The LDA model, an instance of such, is fundamentally probabilistic and proficient at extracting concealed topics within text. This model posits that text data are composed of a three-layer Bayesian probability structure, namely, the document collection layer, the topic layer, and the feature word layer [32]. Fig 2 illustrates this structural arrangement compellingly.

The process of generating "topic-item" of a document is shown in Fig 3 [33], where the white circle is the hidden variable, the green circle is the observed variable, and the rectangle represents the repetition of the variable. The variables are defined in **Table 3** [34].

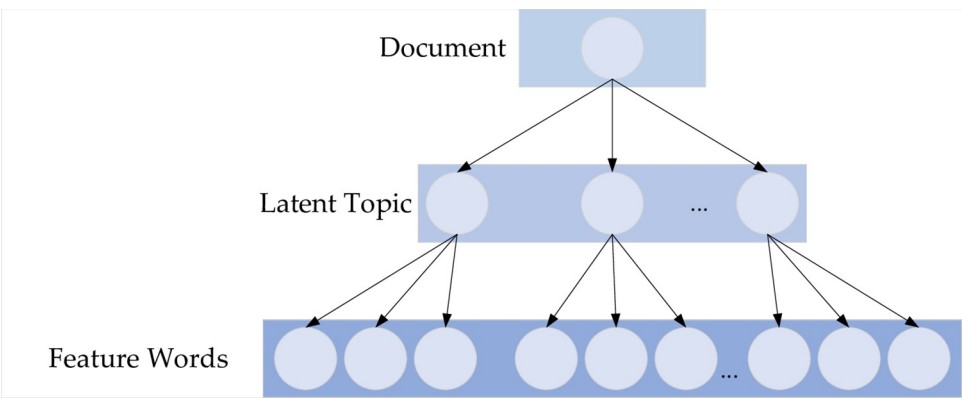

**Fig 2. A three-layer Bayesian probability structure model.**

For a collection of documents $M$, each document has $W$ words that belong to a $N$ set, and each topic belongs to a $K$ set, a set that contains all topics. The generative process occurs in the following manner, a word from $W$, is sampled (with a $\varphi_k \sim$ Dirichlet($\beta$) distribution) and a topic (with a $\theta_m \sim$ Dirichlet($\alpha$) distribution) is assigned to the word, via a $Z$ multinomial distribution of $\theta_m$, to a topic. This generative process is repeated until a convergence of words and topics occurs or another stopping criterion is achieved, for example, a limit of time [35]. The hyperparameters $\alpha$ and $\beta$ control, respectively, the topics distribution over documents and words distribution over topics.

The LDA generative process is used to find the topic allocation for each word, that is to find the subsequent distribution:

◆ Step 1: create distributions $\theta_i \sim \mathrm{Dir}(\alpha)$, where $i \in \{1,\ldots, M\}$ and $\alpha$ is normally dispersed ($\alpha < 1$).

◆ Step 2: create distribution $\varphi_k \sim \mathrm{Dir}(\beta)$, where $k \in \{1,\ldots, K\}$ and $\beta$ is normally dispersed ($\beta < 1$).

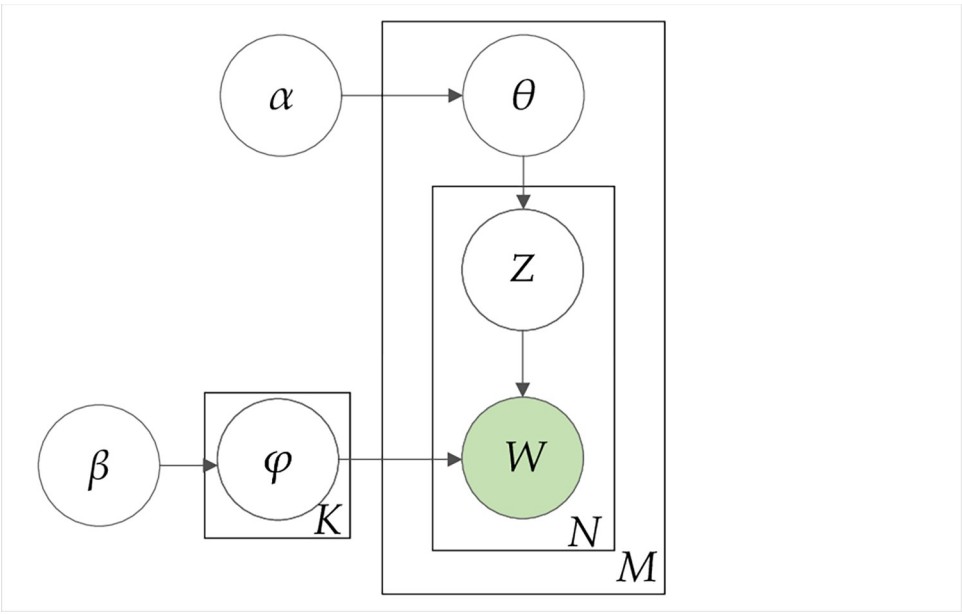

**Fig 3. LDA theme model.**

**Table 3. LDA variables and parameters.**

| Variable | Description |
|---|---|
| $K$ | Represents the number of topics |
| $M$ | Represents the number of documents (corpus) |
| $N_i$ | Represents the number of words ($N$) in each $i$th document |
| $n_{dj}$ | Number of words in a document $d_j$, where $1 \leq j \leq m$ |
| $\alpha$ | Represents a previously defined weight for a Dirichlet distribution relative to a topic $K$ |
| $\beta$ | Represents a previously defined weight for a Dirichlet distribution relative to a word $w$ in a topic |
| $\theta_i$ | Represents the distribution of topics in documents $i$ |
| $\varphi_k$ | Represents a distribution of words in a topic $K$ |
| $z_{ij}$ | Represents the topic assignment for $w_{ij}$ (multinomial distribution) |
| $w_{ij}$ | Represents the $j$th word in the $i$th document (multinomial distribution) |

◆ Step 3: for each word in position $i$, $j$, where $j \in \{1,\ldots,N\}$, and $i \in \{1,\ldots,M\}$:

- —Step 3.1: randomly choose a topic $z_{ij} \sim$ Multinomial($\theta_i$).

- —Step 3.2: randomly choose a word $w_{ij} \sim$ Multinomial($\varphi_k$).

The random variables can be described by the following mathematical notation:
$\varphi_{k\,=\,1\ldots K} \sim$ Dirichlet ($\beta$),
$\theta_{i\,=\,1\ldots M} \sim$ Dirichlet($\alpha$),
$Z_{i\,=\,1\ldots M,\,j\,=\,1\ldots N} \sim$ Multinomial ($\theta_i$),
$W_{i\,=\,1\ldots M,\,j\,=\,1\ldots N} \sim$ Multinomial ($\varphi_k$)

Based on the generative process and observing the dependent relationship between the model's variables, it is possible to describe the probability of all the model's latent variables, given the a priori information. Transcribing these probabilities results in the following joint distribution:

$$P(\varphi, \theta, z, w) = \prod_{k=1}^{K} P(\varphi_k|\beta) \prod_{i=1}^{M} P(\Theta_i|\alpha) \prod_{j=1}^{N} P(z_{ij} = k|w_{d,n}|\theta_i) P(w_{ij}|\varphi_k, z_{ij} = K) \quad (1)$$

or:

$$P(\varphi, \theta, z, w) = \prod_{k=1}^{K} Dir(\varphi_k|\beta) \prod_{i=1}^{M} Dir(\Theta_i|\alpha) \prod_{j=1}^{N} Mult(z_{ij} = k|w_{d,n}|\theta_i) Mult(w_{ij}|\varphi_k, z_{ij} = K) \quad (2)$$

where:

$$\mathrm{Dir}\left(\overrightarrow{\alpha}\right) \rightarrow P\left(\theta|\overrightarrow{\alpha}\right) = \frac{\Gamma(\sum_{i=1}^{k} \alpha_i)}{\prod_{k=1}^{K} \Gamma(\alpha_i)} \prod_{k=1}^{K} \Theta^{\alpha_i - 1} \quad (3)$$

## 3.4 Perplexity evaluation

The evaluation of the LDA topic model typically relies on a metric known as perplexity, utilized to gauge model performance within language modeling; generally, a smaller perplexity index indicates superior model performance [36]. However, solely relying on perplexity to determine the number of topics could potentially lead to model overfitting. Consequently, our attention is not merely anchored on the inflection points within the perplexity-based topic number curve; we also incorporate the visualization effects of the pyLDAvis tool to ascertain the optimal topic number. Subsequently, employing the LDA topic model, we extract the distribution of EC risk topics from two types of evidence sources separately. The formula for

calculating perplexity is as follows:

$$Perplexity = exp\left(-\frac{\sum_M \sum_i^{N_m} lnp(w_{m,i})}{\sum_m N_m}\right) \qquad (4)$$

Where $w_m$ is related to words in document $m$, $N_m$ is the length of document $m$. The better generalization performance is indicated by a lower perplexity over a held-out document.

## 4. Model results

We use the LDA model super-parameter estimation method, which is implemented using the Gibbs sampling method. Gibbs sampling is a Markov Chain Monte Carlo method that is particularly suitable for LDA. It simplifies the sampling process by updating only one variable at each step. This method has proven to be excellent in handling large-scale textual data due to its effectiveness and simplicity [33]. The hyperparameters α and β are set as symmetric Dirichlet prior parameters, where α = 0.1 and β = 0.01, respectively, and the maximum number of iterations is set to 1000. The important parameter of the LDA model is the number of potential topics, and the optimal range of topics for LDA was determined by calculating the perplexity level. The visualization tool pyLDAvis software was used to determine the optimal number of topics. The trend of data perplexity with the number of topics in the text of laws and regulations and the text of adjudication documents is shown in Figs 4 and 5, and the visualization of the distribution of topics drawn using the visualization tool pyLDAvis software is shown in Figs 6 and 7, respectively.

In Figs 4 and 5, the horizontal coordinate indicates the number of topics, and the vertical coordinate perplexity indicates the magnitude of perplexity. In Figs 6 and 7, the circles represent different themes; the size of the circles represents the probability of the occurrence of each theme; and the distance between the circles represents the degree of correlation between the themes; a smaller degree of overlap between the circles indicates that the theme division is more effective. As observed in Fig 4, the minimum perplexity is achieved when the number of topics is approximately 15. Combining this insight with Fig 6 allows us to determine the optimal number of topics for class A evidence, which is 4. Similarly, Fig 5 shows a gradual decrease in perplexity with an increasing number of topics, with the curve's inflection point falling

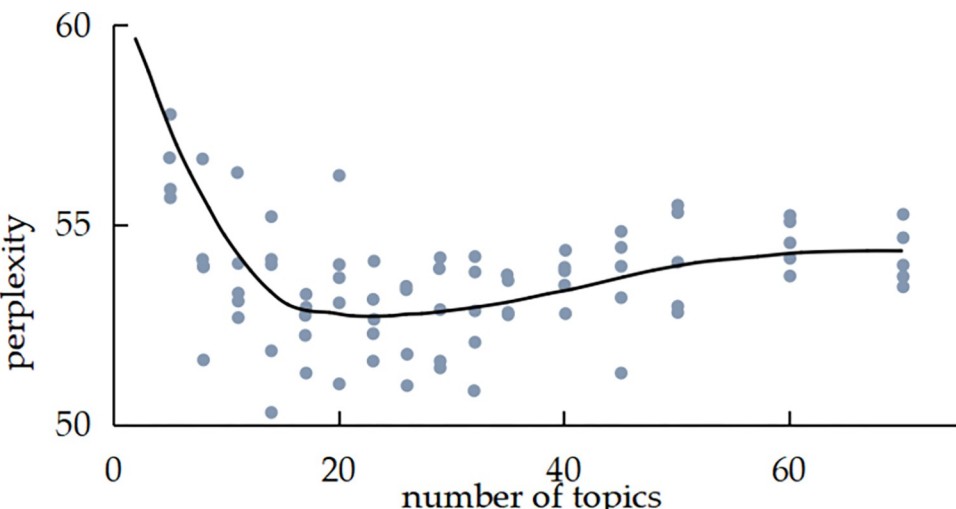

**Fig 4. Perplexity index of class A evidence.**

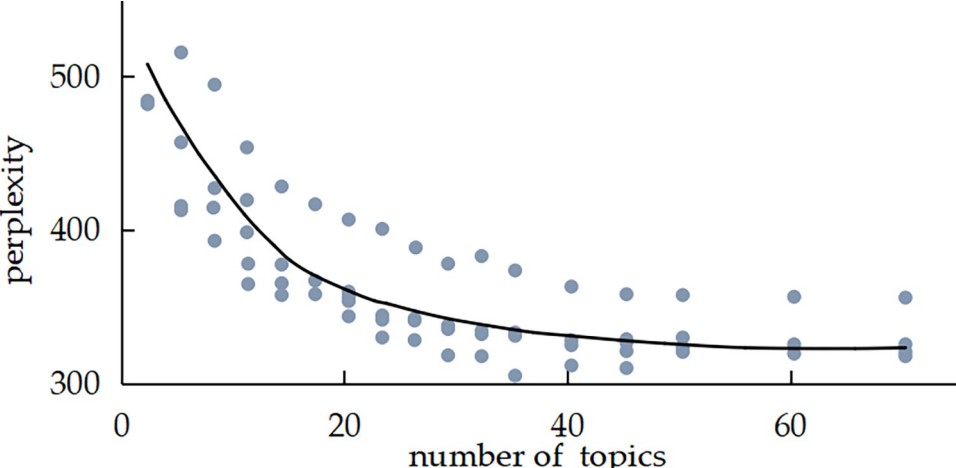

**Fig 5. Perplexity index of class B evidence.**

within the range of [5, 20]. By combining this information with Fig 7, we can determine the optimal number of topics for class B evidence in Judgment documents, which is 5.

We obtained the keyword term distribution by classifying the LDA topic model. We sorted out the top 10 high-probability feature words corresponding to the topic to create the keyword distribution tables for the topic of EC risk, as shown in Tables 4 and 5.

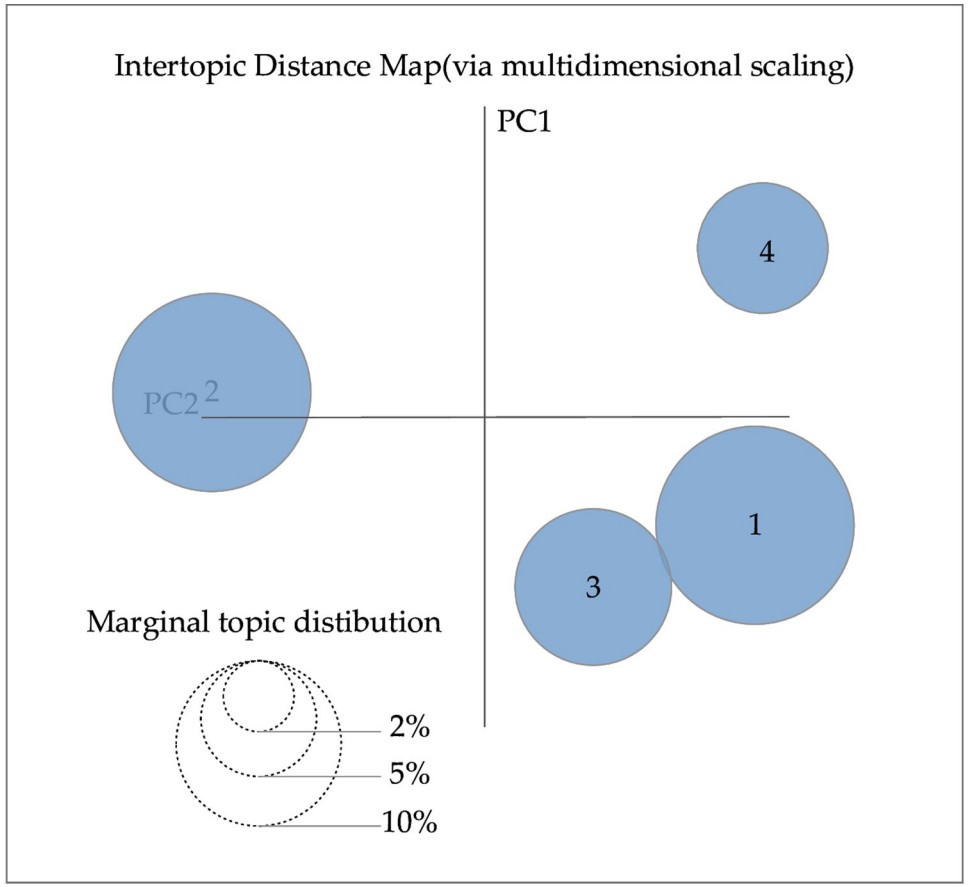

**Fig 6. Visualization of theme extraction results from class A evidence.**

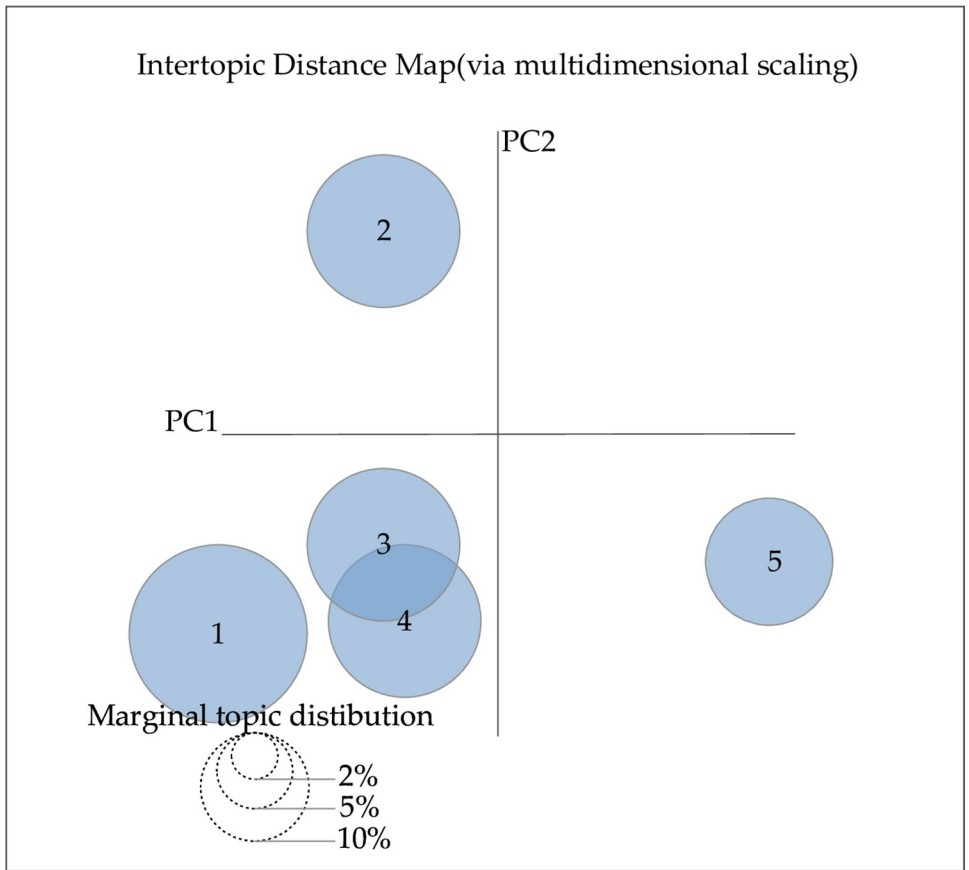

**Fig 7. The visualization of topic extraction results from class B evidence.**

As shown in Table 4, evidence in class A is categorized into four main themes. Firstly, topic A keywords predominantly center around bidding. Issues such as opacity and information inaccuracy during the bidding phase can often lead to ECs. This is frequently caused by ambiguity in requirements and incomplete bidding documentation. Topic B keywords are primarily related to contracts and financial management. Factors such as budget overruns and unclear contractual terms often result in financial disputes and schedule delays, which can ultimately trigger project changes. Moving on to Topic C, it revolves around design and project changes. These changes come with their own set of risks, including

**Table 4. Key keywords for topic modeling of class A evidence.**

| Topic Number | keywords |
|---|---|
| Topic A | Bidding, Project, Bid Inviter, Bidder, Winning Bid, Bid Evaluation, Bidding Documents, Bid, Responsibility, Bid Documents |
| Topic B | Contract Awarder, Construction Project, Contractor, Contract, Engineering, Guarantee Money, Price Money, Request, Agreement, Deadline data |
| Topic C | Design, Change, Engineering, Alteration, [Verb], Project, Document, Building, Modification, [Noun], Legal Entity, Plan |
| Topic D | Engineering, Construction, Contracting, Awarding, Work, Contract, Qualification, Subcontracting, Construction Work, Survey |

**Table 5. Key keywords for topic modeling of class B evidence: Judicial documents.**

| Topic Number | keywords |
|---|---|
| **Topic 1** | Construction Project, Construction Contract, Fulfillment, Project Cost, Delivery, Agreement, Settlement, Error, Termination, Interest Rate |
| **Topic 2** | Design, Engineering Construction, Drawings, Winning Bid, Bidding, Visa Form, Plan, Bidding Documents, Management, Construction Unit |
| **Topic 3** | Settlement, Materials, Amount, Data, Unit Price, Adjustment, Equipment, Bill of Quantities, Consultation, Project Owner |
| **Topic 4** | Project Cost, Amount, Increase, Site, Visa, Unit Price, Adjustment, Pricing, Drawings, Change |
| **Topic 5** | Contractor, Contract Awarder, Party B, Party A, Breach of Contract, Completion and Acceptance, Progress Payment, Construction Period, Contract Price, Quality |

uncertainties during the design stage and their potential ripple effects during implementation. Finally, Topic D highlights construction and quality management. It's essential to consider factors like subcontractors' control over construction quality and schedules, as well as the technical challenges that may arise during construction, all of which can be significant triggers for ECs.

Table 5 displays the categorization of class B evidence in arbitration documents concerning ECs into five topics. In Topic 1, the focus is on risks related to financial and contract performance. Keywords like '工程价款' (engineering fund), '解除' (termination), and '利率' (interest rate) reveal that fluctuations in investment or contract terminations can potentially lead to cost increases and project delays. Topic 2 sharpens the focus on risks in the design phase and the initial tendering process. Keywords like '' (design), '图纸' (blueprints), and '招投标' (bidding) indicate that inaccuracies in design or unfairness in the tendering process may result in conflicts, rework, and schedule alterations during implementation. Topic 3, reflecting on '材料' (materials), '金额' (amount), and '调整' (adjustment), highlights risk in materials management and cost adjustment. Changes in materials or adjustments in the engineering volume can impact the project's budget and timeline. Entering Topic 4, keywords such as '工程造价' (engineering cost), '签证' (visa), and '变更' (change) unveil risks in on-site management. Instability in on-site operations, including uncertainties in engineering costs, unplanned changes, and on-site visas, can lead to increased time and costs. Lastly, Topic 5 accentuates risks in contractual relationships and delivery, involving elements like '承包人' (contractor), '发包人' (employer), and '合同价款' (contract fund). Disharmony in cooperation between parties or delivery delays may trigger contract disputes or changes. In summary, these topics distinctly highlight the multifaceted EC risks in engineering projects, presenting a comprehensive challenge to project management teams across contracts, finances, design, materials management, on-site management, and personnel.

## 5. Evidence fusion and discussion

### 5.1 Evidence fusion

Evidence fusion serves to enhance the comprehensive perspective of risk management, thereby bolstering the quality of decision-making. By integrating Type A and Type B evidence, we can guide decisions in a more scientific manner. Specifically, as depicted in Fig 8, there is a notable correlation between Type A evidence in regulatory standard text and Type B evidence in judicial documents across various topics. For instance, Topics B and D are intimately related to Topic 1, which encompasses the execution of construction contracts, the selection of contract types, and the settlement of construction funds. Moreover, Topic A, which focuses on bidding,

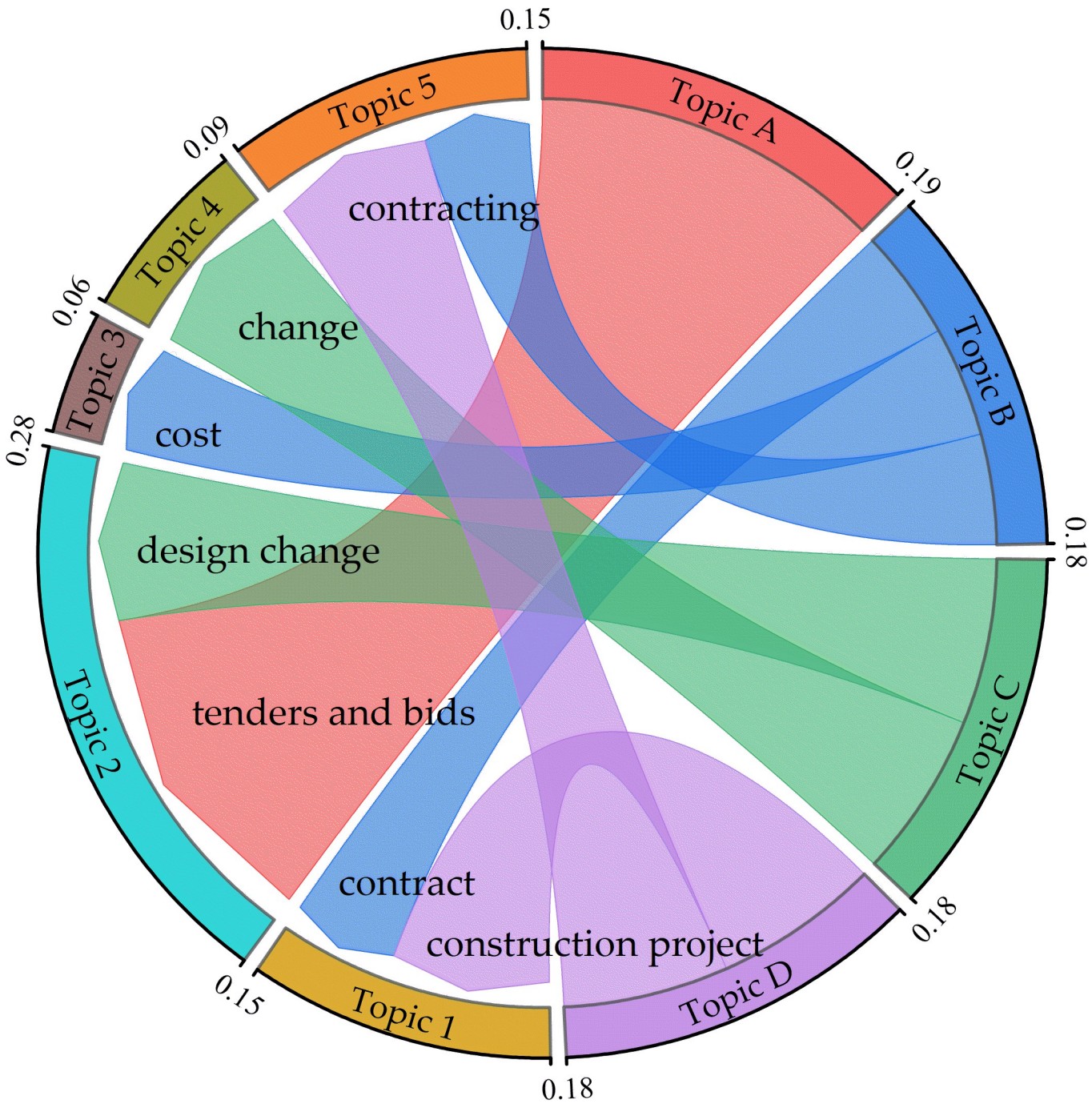

**Fig 8. Relationship of themes between type A evidence and type B evidence.**

and Topic C, which pertains to ECs, are closely aligned with Topic 2. Furthermore, Topic C correlates with Topic 4 in terms of ECs and adjustments to construction funds. Lastly, Topics B and D are significantly connected to Topic 5, which revolves around relationships in subcontracting and themes of contract funds. To quantify the relationships among these themes, we

employ the correlation coefficient calculation method presented in Eq (5).

$$\eta = \frac{n}{k_1 N_1 + k_2 N_2} \tag{5}$$

In Eq (5), the symbol $\eta$ represents the interrelation coefficient among EC risk themes at different evidence levels. Here, $n$ is defined as the count of identical or similar keywords present in regulatory standard text topics and adjudication document topics. $K_1$ and $K_2$ designate coefficients for class A evidence topics and class B evidence topics, while $N_1$ and $N_2$ represent the total keyword count within class A and class B evidence topics, respectively. In this model, $K_1$ and $K_2$ are assigned values of 0.6 and 0.4, respectively. A threshold of $\eta > 0.03$ is set to determine theme fusion. The results of the theme integration are shown in Fig 8.

## 5.2 Evaluation of evidence

The text of ECs involves specialized knowledge in the field of construction, which needs to be fully integrated with the objective evidence of text mining to analyze the evidence of category C expert opinions. Therefore, according to the evidence classification table, the subject factors of EC risk are extracted to design the questionnaire by integrating the standard text of class A evidence regulations and the subject of class B evidence judgment documents.

This study was approved by the ethics committee of the Modern Project Management Team of the College of Hydraulic and Environmental Engineering, China Three Gorges University. We certify that the study was performed in accordance with the 1964 declaration of HELSINKI and later amendments. In addition, written informed consent was obtained from all the participants prior to the enrollment (or for the publication) of this study. The questionnaire recruitment in this study started on May 05, 2023 and ended on May 20, 2023.

The questionnaire survey sent the link to fill in the questionnaire to 26 experts in the construction industry 3 professors majoring in engineering management, 10 senior engineers, 4 registered first-class builders, 2 registered first-class architects, and 7 registered second-class builders through WeChat, etc. The questionnaire survey mainly covers the provinces of Hubei, Hunan, Guangdong, and Sichuan in China. This time, the interviewees have a certain amount of professional knowledge and work experience in the engineering field, thus guaranteeing the professionalism and accuracy of the questionnaire data.

We collected 26 valid questionnaires and performed reliability analysis on the survey results using SPSS. Notably, with a Cronbach's coefficient of 0.958, exceeding 0.8, the data showcases exemplary reliability. In addition, we had experts assess the probability of occurrence $P$ of EC risks on a scale of 0.2, 0.4, 0.6, 0.8, and 1, as well as the risk's hazard degree $C$ on a scale of 1–5. To comprehensively evaluate the risk's critical degree, we calculated an average score for both $P$ and $C$. Herein, the $PC$ value is the product of the average scores of the two, with the calculation process shown in Formula (6). A larger $PC$ value signifies more critical risk, and the detailed assessment results can be seen in Table 5.

$$PC = \bar{P} \times \bar{C} = \frac{\sum P_m \times \sum C_m}{m^2} \tag{6}$$

In Eq (6), $\bar{P}$ represents the average score for the probability of risk occurrence., $\bar{C}$ is the average score for the risk severity, $P$ is the probability of EC risk occurrence, $C$ stands for risk severity, and $m$ is the number of scores.

As shown in Table 6, the conclusions of the EC risk analysis are as follows:

**Table 6. EC risk assessment results.**

| Topic | Risks | Probability of Risk Occurrence $\bar{P}$ | Risk Criticality $\bar{C}$ | PC |
|---|---|---|---|---|
| Topic I Contract | Improper selection of construction contract type | 0.546 | 2.730 | 1.491 |
| | No construction qualification, invalid contract | 0.623 | 3.115 | 1.941 |
| | Contract claim | 0.646 | 3.231 | 2.088 |
| | Omission or misrepresentation of contract terms | 0.653 | 3.269 | 2.138 |
| Topic II Technique | Design defects, errors, and omissions | 0.753 | 3.769 | 2.841 |
| | Failure to consider construction feasibility in design | 0.700 | 3.500 | 2.45 |
| | Engineering survey quality | 0.700 | 3.500 | 2.45 |
| | Unreasonable construction technology and scheme | 0.661 | 3.308 | 2.188 |
| | Inappropriate construction safety measures | 0.669 | 3.538 | 2.368 |
| | Inadequate consideration of site conditions | 0.546 | 2.731 | 1.491 |
| Topic III Fund | Lack or shortage of funds | 0.669 | 3.346 | 2.239 |
| | Inflation | 0.554 | 2.769 | 1.534 |
| | Supply and demand risk of main material equipment market | 0.615 | 3.076 | 1.893 |
| Topic IV Personnel | Poor quality or weak operational capability among participants. | 0.662 | 3.308 | 2.188 |
| | Incoordination within and among participating parties | 0.631 | 3.154 | 1.989 |
| Topic V Others | Irresistible force of nature | 0.562 | 2.808 | 1.577 |
| | Complex engineering geological conditions | 0.608 | 3.038 | 1.846 |
| | Impact of construction on the environment | 0.600 | 3.000 | 1.800 |
| | Policy adjustment | 0.582 | 2.923 | 1.709 |
| | Mandatory provisions adjustment of design and construction specifications | 0.608 | 3.038 | 1.846 |
| | Improper selection of materials and equipment | 0.569 | 2.846 | 1.620 |
| | Improper installation and use of materials and equipment | 0.577 | 2.885 | 1.664 |

1. The *PC* value of design defects, errors, and omissions is the largest, which is the key risk of ECs.

2. The *PC* value of 2.45 for the factors of design without considering the possibility of construction and the quality of the engineering survey is an important risk for EC.

3. Other risk factors, such as construction safety, allocation of funds, rationality of technological approaches, personnel quality, and clarity of contract clauses, which all bear *PC* values above 2.0, exert substantial impact.

## 5.3 Discussion

Navigating the intricacies of EC risk management, this study ventures beyond the confines of traditional evidence-based management practices by methodically categorizing EC risk evidence into three distinct levels: regulatory standards (class A), scholarly research (class B), and professional insights (class C) [37]. This layered framework not only deepens our comprehension of EC risks but also highlights the varied influence of evidence types on the appraisal and management of risks.

Central to our findings is the identification of design defects, errors, and omissions, underscored by their leading PC values, as pivotal elements necessitating focused intervention within EC management strategies. This insight corroborates the widely accepted notion that lapses during the design phase are significant contributors to project vulnerabilities, underscoring the imperative for rigorous design verification processes [38]. Furthermore, the notable PC value linked to neglecting construction feasibility and the integrity of engineering surveys

underscores the criticality of embedding exhaustive feasibility analyses at the outset of the design phase.

Our investigation also brings to light that risk elements traditionally viewed as peripheral—such as safety protocols, financial planning, and contractual clarity—exert a pronounced impact on project outcomes, as evidenced by their PC values exceeding the 2.0 benchmark. This revelation prompts a reevaluation of risk prioritization in EC management, advocating for a more comprehensive approach that balances technical and administrative risk considerations.

Employing expert evaluations to determine the likelihood (P) and severity (C) of EC risks introduces a layered complexity to our analysis. While expert input enriches our study with invaluable insights, it also beckons a consideration of the subjective nature of risk assessments and the variability inherent in expert judgments. This highlights the necessity for establishing standardized protocols for expert contributions, aiming to enhance the reliability and impartiality of risk evaluations.

Furthermore, the adoption of evidence grading theory within our analytical model provides a structured approach to deciphering the hierarchy of evidence, enabling a more organized evaluation of EC risks. This innovative integration not only bridges the theoretical and practical aspects of risk management but also pioneers avenues for subsequent research endeavors in this domain.

Nevertheless, the study acknowledges its limitations. The stratification of evidence sources, though enlightening, might not fully encapsulate the multifaceted dynamics influencing EC risks. Future inquiries should strive to refine these categories, perhaps introducing more nuanced subdivisions to more accurately portray the diverse spectrum of evidence affecting EC risk management.

In sum, our investigation highlights the complex nature of EC risks and underscores the necessity for a holistic risk management paradigm that integrates technical, administrative, and experiential perspectives. By promoting the application of evidence grading theory and spotlighting the significance of oft-overlooked risk factors, we contribute to a more comprehensive understanding of EC risk management. This endeavor lays the groundwork for the development of more refined and effective risk mitigation strategies in subsequent engineering endeavors.

To address the study's findings, we propose several actionable recommendations aimed at bolstering EC risk management:

**1. Enhancing design quality:**

Owners are advised to strengthen the management of surveying and design units to improve design depth and quality.

Construction units should thoroughly analyze construction drawings before commencing on-site work to avoid discrepancies.

**2. Cutting-edge technologies and standards:**

Emphasize the use of cutting-edge technologies and standards to ensure the innovative and accurate nature of design schemes.

**3. Continual communication and timely adjustments:**

Stress the importance of continual communication and timely adjustments to ensure the smooth implementation of design plans.

**4. Thorough on-site survey during construction:**

Advocate for a comprehensive on-site survey during construction, especially focusing on geological conditions.

**5. Constructability and adherence to standards:**

Ensure the constructability of designs and their adherence to standards based on survey reports during the design phase to prevent major design alterations later.

**6. Strict adherence to change management procedures:**

Recommend strict adherence to change management procedures and enhanced foresight in every preliminary phase to mitigate risks.

**7. Financial control and contract analysis:**

Understand the investor's financial strength, exercise stringent control over the use of special funds, and conduct a thorough analysis of contract details.

**8. Specialized risk management team:**

Consider establishing a specialized risk management team to conduct regular risk assessments and adjust coping strategies for risks at different stages.

## 6. Conclusions

By fusing the text of class A regulatory standards with the themes of Level B judicial documents, the study not only refines thematic factors of EC risk but also crafts survey questionnaires, plunging into an in-depth analysis of Level C evidence, which encapsulates expert opinions.

Identifying predominant risks in ECs through LDA topics the findings reveal that the LDA topics related to EC risks encompass five main categories: contract risks, technical risks, fund risks, personnel risks, and various other risks. Upon a comprehensive analysis, it is evident that technical risks stand out as the most critical.

The findings suggest that the integration of text mining methods with evidence grading theory enables a scientific and effective analysis of EC risk topics in construction projects. This approach facilitates the identification of key risks, ensuring the highest levels of scientific rigor and reliability in the research.

Future research endeavors should delve deeper into refining the selection of topic models during data processing, exploring advanced text mining techniques, and broadening the range of high-level evidence collection to formulate a more comprehensive risk analysis model. Furthermore, there is a need to validate the research outcomes in real engineering projects and evaluate the practical utility of the model. This study by integrating evidence grading theory and text mining methods, presenting an innovative approach to EC risk analysis. In practical terms, our research provides valuable guidance for risk management in engineering projects, particularly during the design and construction phases. On the policy front, our study establishes a groundwork for the formulation of relevant policies to ensure the scientific management and risk control of ECs.

## Author Contributions

**Conceptualization:** Lianghai Jin, Chenxi Li, Songxiang Zou.

**Data curation:** Chenxi Li.

**Formal analysis:** Lianghai Jin, Chenxi Li.

**Funding acquisition:** Lianghai Jin, Zhongrong Zhu.

**Investigation:** Lianghai Jin, Chenxi Li, Zhongrong Zhu, Songxiang Zou, Xushu Sun.

**Methodology:** Lianghai Jin, Chenxi Li, Zhongrong Zhu.

**Project administration:** Lianghai Jin, Chenxi Li, Zhongrong Zhu, Songxiang Zou, Xushu Sun.

**Resources:** Lianghai Jin.

**Software:** Lianghai Jin, Chenxi Li.

**Supervision:** Lianghai Jin, Zhongrong Zhu, Xushu Sun.

**Validation:** Lianghai Jin, Chenxi Li.

**Visualization:** Lianghai Jin, Chenxi Li.

**Writing – original draft:** Lianghai Jin, Chenxi Li.

**Writing – review & editing:** Lianghai Jin, Chenxi Li.

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
