## [Decision Letter · Decision Letter 0]

20 Dec 2023

PONE-D-23-35817Mining LDA topics for graded evidence on construction engineering change risks: a Chinese data analysisPLOS ONE

Dear Dr. li,

Thank you for submitting your manuscript to PLOS ONE. After careful consideration, we feel that it has merit but does not fully meet PLOS ONE’s publication criteria as it currently stands. Therefore, we invite you to submit a revised version of the manuscript that addresses the points raised during the review process.

We look forward to receiving your revised manuscript.

Kind regards,

Saliha Karadayi-Usta, PhD

Academic Editor

PLOS ONE

Journal Requirements:

Department of Humanities and Social Affairs Fund of the Ministry of Education ( 21YJA630038).

5. Please amend the manuscript submission data (via Edit Submission) to include author Dr. Lianghai Jin.

Reviewers' comments:

Reviewer's Responses to Questions

**Comments to the Author**

1. Is the manuscript technically sound, and do the data support the conclusions?

Reviewer #1: No

Reviewer #2: Partly

Reviewer #3: Yes

2. Has the statistical analysis been performed appropriately and rigorously? 

Reviewer #1: No

Reviewer #2: N/A

Reviewer #3: Yes

3. Have the authors made all data underlying the findings in their manuscript fully available?

Reviewer #1: No

Reviewer #2: Yes

Reviewer #3: Yes

4. Is the manuscript presented in an intelligible fashion and written in standard English?

Reviewer #1: No

Reviewer #2: No

Reviewer #3: No

5. Review Comments to the Author

Reviewer #1: Title: graded evidence? What's this? Remove Chinese from the title.

Too many abbreviations, please use full name except for common name like LDA.

Something like probability of risk occurrence P and the risk importance C and calculate the average product PC of the two to determine the key risks are not needed.

synthesizing information from both class A and B evidence, not sure what is this.

a total of

five prominent risk themes were identified, namely contract, technology, funds, personnel, and other hazards. These are known factors for long?

What is EC regulatory standards? Outsiders have no way to know what's this.

shorten 2.1 and do not use abbreviation like ecs

Classifying evidence according to its credibility is crucial for its scientific and pragmatic application in risk management, particularly in the sphere of EC. ? By what means?

Line 96-104, missing citation.

The whole page where Figure 2 locates have missing citaiton.

State what is LDA model etc when it firstly appeared in introduction and research method, the LDA section completely missed citation: Public Opinion Mining on Construction Health and Safety: Latent Dirichlet Allocation Approach, L Zeng,et al Buildings, 2023

Results: where α = 0.1 and β = 0.01, what do these values mean?

How was the data obtained for LDA, what are the selection and exclusion criteria?

State where has survey been used: Ranking of risks for existing and new building works, Sustainability, 2019

The questionnaire survey sent the link to fill in the questionnaire to 26 experts? Why 26? What are criteria of experts?

Add discussion section.

What is the model for 26 surveys?

Extend conclusion, limitation, direction of future research, academic, practical and policy contributions.

Reviewer #2: Dear authors, the research proposal is interesting, but I think the authors could improve the text and structure of the article before we consider it for publication. I would like to make a few suggestions:

1) In the introduction, the authors did not explicitly state the problem question or the research objectives;

2) In the materials and methods section, the authors talk about data collection, but they don't mention when the data was collected, how it was collected and in which database. I would like these questions to be clarified.

3) The authors used LDA. This technique is excellent for large databases. The authors have described it, but they could provide examples of the use of LDA and a full description of the method's formulation. I suggest considering the following applications of LDA: https://doi.org/10.1108/JM2-10-2020-0268;
https://doi.org/10.1108/DTA-12-2018-0109.

4) The authors could put together a table with all the parameters they used in LDA, such as the number of topics, and how they defined the number of topics.

Reviewer #3: This paper is written clearly, and it is worth publishing, but before publication, the following issues should be considered:

1- The abstract: Writing it more concisely and in a richer form is better. The end of the abstract should discuss the datasets and results.

2- Keywords: could be more precise. Please recheck this section and write LDA in full format (Latent Dirichlet Allocation).

3- Introduction: Please write the main contributions of the research at the end of the Introduction. Please extract the literature review from the Introduction and write it in a separate section after the Introduction section with the name of the related works section.

4- The part of the related works must be completed and rewritten by recent articles from reputable journals.

5- Please consider an example from the real world to describe your method. On the other hand, this method can be better shown to the readers by a flow chart.

6- Why did you use the Gibbs sampling method? Please explain it in the article

7- Please explain more about how you evaluated your work by the formula. For example, how did you evaluate your data by this criterion? On what part of the data did you do this? And how? Please explain this more accurately and completely.

8- Please also compare your work with the methods presented recently.

9- In the evaluation section, please explain wholly and accurately under what conditions, on what machines, and under what parameter values you compare the methods. Please mention it in some tables. It is necessary to reproduce the data by the readers of this section. Also, in this section, mention the complete information about what kind of datasets you used and the information of those datasets, as well as training data, test data, and all evaluation conditions.

10- One of the fundamental questions is, what are the advantages and disadvantages of your method compared to the recent methods?

11- There are some other works on semantic LDA which is not considered in this paper, such as:

1-https://doi.org/10.1007/s00180-022-01246-z

2-https://doi.org/10.1155/2022/7612276

3-https://doi.org/10.1111/exsy.12527

4-https://doi.org/10.1142/S0219622022500584

5-https://doi.org/10.1016/j.neucom.2022.10.002

6. PLOS authors have the option to publish the peer review history of their article (what does this mean?). If published, this will include your full peer review and any attached files.

Reviewer #1: No

Reviewer #2: No

Reviewer #3: No

---

## [Author Response · Author response to Decision Letter 0]

29 Feb 2024

Response to Reviewers

Dear editor and reviewers,

We gratefully appreciate the editors and all reviewers for their time spend making positive and constructive comments. These comments are all valuable and helpful for revising and improving our manuscript entitled "Mining LDA topics for graded evidence on construction engineering change risks: a Chinese data analysis (Manuscript Number: PONE-D-23-35817, as well as the important guiding significance to our research. 

We have studied comments carefully and have made correction which we hope meet with approval. Revised portions are marked in red in the revised manuscript. The summary of corrections and the responses to the reviewer's comments are listed in the Revision Report.

Thank you and best regards.

Yours sincerely

Lianghai Jin

Corresponding author

Email: jinlianghai@ctgu.edu.cn

Revision Report

First of all, I would like to express our sincere gratitude to the reviewers for their comments. These comments are all valuable and helpful for revising and improving our manuscript, as well as the important guiding significance to our researches. We have studied comments carefully and have made correction which we hope to meet with approval. Revised portions are marked in red in the revised version. The summary of corrections and the responses to the reviewer's comments are listed below.

Summary of the revision:

Section 0&1: We have explicitly stated the problem question and research objectives in the introduction and improved the abstract for conciseness and included a discussion of datasets and results. 

Section 2 (Related works): We have added a new section, Section 2: Related Works.

Section 3 (Method): We have Provided a flowchart to illustrate the methodology (Fig1). In addition, we have incorporated additional citations in the methodology section to enhance the referencing of our research. Furthermore, we have provided detailed explanations regarding the data sources (Table2), parameter settings (Table3), formula interpretations, and the modeling process.

Section 4 (Model Results): We explained what Gibbs sampling is and the rationale behind choosing Gibbs sampling in our research.

Section 5 (Evidence fusion and discussion): We revised the managerial recommendations and added more interesting and promising discussions by the comments and suggestions from reviewers.

Section 6 (Conclusion): We supplemented information about the timing of the questionnaire survey and other details and added a discussion section.

In addition, the manuscript has been carefully revised considering the language and grammar problems, including the sentences that are convoluted and hard to follow.

Responses to reviewers 

(Original comments by reviewers are in blue color)

Reviewer # 1:

1. Comment: Graded evidence? What's this? 

1. Reply: Thank you for your question. Graded evidence refers to the application of the theory of graded evidence to grade the reliability of evidence in the area of engineering changes risk. We appreciate your valuable advice once again and will strive to improve the paper.

2. Comment: Remove Chinese from the title.

2. Reply: Thank you for your advice. We have revised the title to accurately and engagingly reflect it contend. We appreciate your valuable advice once again and will strive to improve the paper.

For example:

original title: Mining LDA topics for graded evidence on construction engineering change risks: a Chinese data analysis 

revised title: Mining LDA topics for graded evidence on construction engineering change risks: a data analysis 

3. Comment: Too many abbreviations, please use full name except for common name like LDA.

3. Reply: Thank you for your advice. We have carefully considered your suggestions regarding the use of abbreviations. Abbreviations with a frequency of less than 3 in the text have been removed to enhance the readability and clarity of the paper. This decision was made to better meet the expectations of our readers and to make our research more accessible and understandable. We appreciate your valuable advice once again and will strive to improve the paper.

For example: (Line48)

Original word：

evidence-based management (EBM)

Revised word: 

evidence-based management

4. Comment: Something like probability of risk occurrence P and the risk importance C and calculate the average product PC of the two to determine the key risks are not needed.

4. Reply: Thank you for your advice. We've revised the abstract and removed this passage from the text. Your guidance has been instrumental, and we appreciate your thorough review.

Original sentence：

In addition, the questionnaire method was used to determine the probability of risk occurrence and the risk importance C and calculate the average product PC of the two to determine the key risks. 

Revised sentence: 

 In addition, by combining EC risk topics with relevant literature, we identified factors influencing EC risks. Subsequently, we designed an expert survey questionnaire to determine the key risks and important risk topics associated with potential risks.

5. Comment: Synthesizing information from both class A and B evidence, not sure what is this.

5. Reply: Thank you for your comment. The mention of "Class A and B evidence" refers to the integration of information from highly credible sources. Class A represents evidence of the highest quality and reliability, such as information related to engineering change regulations. On the other hand, Class B represents evidence of higher quality or reliability, such as data related to engineering changes. The combination of this evidence grading and fusion contributes to ensuring a comprehensive analysis in our study, taking into account a variety of evidence types.

6. Comment: A total of five prominent risk themes were identified, namely contract, technology, funds, personnel, and other hazards. These are known factors for long?

6. Reply: Thank you for your advice. The identification of five prominent risk themes—contract, technology, funds, personnel, and other hazards—has been a deliberate and comprehensive effort in our study. While these factors are recognized as longstanding considerations, our aim is to provide a nuanced and up-to-date analysis that integrates both historical knowledge and contemporary insights. The inclusion of these well-established factors serves as a foundation for our research, allowing us to build upon existing understanding and explore potential developments or shifts in the dynamics of engineering change risks. We appreciate your valuable advice once again and will strive to improve the paper.

7. Comment: What is EC regulatory standards? Outsiders have no way to know what's this.

7. Reply: Thank you for raising this question. 'EC regulatory standards' refer to the legal regulations and industry standards associated with engineering changes. To enhance clarity for readers who may not be familiar with this term, I have provided a brief explanation in the abstract. This will aid in ensuring that all readers, including those unfamiliar with the specific terminology, can easily comprehend the background of our research.

Original sentence：

and the LDA topic model was used to analyze the EC regulatory standards and judgment documents to detect EC risk themes.

Revised sentence: 

 This involved analyzing regulations, industry standards, and judgment documents related to EC, ultimately identifying the themes associated with EC risks.

8. Comment: shorten 2.1 and do not use abbreviation like ecs.

8. Reply: Thank you for your advice. We've shortened 2.1 and revised the esc abbreviation. We appreciate your valuable advice once again and will strive to improve the paper.

For example:

original title: Constructing a grading framework for risk evidence in ecs 

revised title: EC risk evidence grading framework 

9. Comment: Classifying evidence according to its credibility is crucial for its scientific and pragmatic application in risk management, particularly in the sphere of EC. ? By what means?

9. Reply: Thank you for your valuable advice. The paper references the work of Haitao Zhang et al. [1] in construction safety management and prevention. Their research focuses on evidence grading, emphasizing the credibility of evidence. They define evidence in both medical and safety management fields as: 

① the basis for determining facts 

② materials used for demonstration

 The study includes a comparative analysis of evidence sources in these domains. 

Conclusively, evidence in construction safety management is classified into three levels based on credibility. Regulatory evidence, such as national laws and industry standards, holds the highest credibility, followed by literature-based evidence, including systematic reviews and practical experiences. Other evidence types encompass statements from safety organizations, expert opinions, recognition from authorities, and conference reports. We appreciate your advice and are committed to enhancing the paper.

1 Zhang H.; Wu W.; Li X.; Li Q. Research on construction site safety accident prevention system based on evidence-based approach. Chin. J. Saf. Sci. 2012, 22, 17–22. DOI: 10.16265/j.cnki.issn1003-3033.2012.12.006.

10. Comment: Line 96-104, missing citation.

10. Reply: Thank you for pointing out the missing citation. Thank you for pointing out the missing citation. We acknowledge this oversight and have added the appropriate citations in the revised version of the manuscript, specifically in previously line 96-104( Now line 169-177). We appreciate your advice and are committed to enhancing the paper.

11. Comment: The whole page where Figure 2 locates have missing citation.

11. Reply: Thank you for bringing the missing citation in the vicinity of Figure 2 to our attention. We have carefully addressed this concern and made the necessary modifications in the revised version of the manuscript. Your diligence in reviewing our work is greatly appreciated. We understand the importance of accurate and complete references and assure you that we are committed to upholding the highest standards in our research. Your valuable advice has contributed to the enhancement of our paper, and we look forward to any further guidance you may provide.(Line 243-346)

12. Comment: State what is LDA model etc when it firstly appeared in introduction and research method, the LDA section completely missed citation: Public Opinion Mining on Construction Health and Safety: Latent Dirichlet Allocation Approach, L Zeng,et al Buildings, 2023

12. Reply: Thank you for your advice. We have provided a more detailed explanation of the LDA model in both the introduction and the LDA model sections. Additionally, we have addressed the absence of citation in the LDA section by including the reference: "Public Opinion Mining on Construction Health and Safety: Latent Dirichlet Allocation Approach, L. Zeng et al., Buildings, 2023." We appreciate your valuable suggestions, and we will continue to strive for clarity and quality in the paper.

13.Comment: Results: where α = 0.1 and β = 0.01, what do these values mean? 

13. Reply: Thank you for your advice. In LDA, each document is considered a mixture of multiple topics, and α is a parameter that controls the distribution of topics in a document. A higher α value leads to documents containing more topics, while a lower α value makes documents more inclined to include fewer topics. The typical range for α is a positive real number. LDA assumes that each topic is composed of multiple words, and β is a parameter that controls the distribution of words in a topic. A higher β value results in topics containing more words, while a lower β value makes topics more inclined to include fewer words. Similar to α, β is usually a positive real number.

These two parameters influence the distribution of topics in documents and the distribution of words in topics during the training process of LDA. Adjusting their values can impact the characteristics of the final topic model obtained. Setting α = 0.1 and β = 0.01 has been found to yield effective results in topic modeling. We appreciate your valuable advice once again and will strive to improve the paper.

14. Comment: State where has survey been used: Ranking of risks for existing and new building works, Sustainability, 2019 

14. Reply: Thank you for your suggestion. We have added the location information in the manuscript. The questionnaire survey primarily covers the provinces of Guangdong, Hunan, Hubei, and Sichuan in China, representing diverse geographical and cultural backgrounds. This selection is aimed at ensuring our research has broad coverage in terms of diversity and representativeness. We have included the citation in the text: "Ranking of risks for existing and new building works, Sustainability, 2019." We appreciate your thorough review and valuable feedback, and we are committed to ensuring the integrity and accuracy of our research. If you have any further suggestions or concerns, please feel free to let us know.

15. Comment: The questionnaire survey sent the link to fill in the questionnaire to 26 experts? Why 26? What are criteria of experts?

 15. Reply: Thank you for your inquiry. The selection of 26 experts was based on specific criteria, considering their expertise and experience in the field of construction and risk assessment. These experts were chosen due to their significant contributions to relevant research, professional background, and knowledge in the subject matter. We aimed to gather insights from a diverse group of experts to ensure a comprehensive and well-rounded understanding of the topic. The criteria for selecting experts included their academic qualifications, industry experience, and prior involvement in construction risk-related projects. We appreciate your attention to this matter and are open to any suggestions or further inquiries.

16. Comment: Add discussion section.

16. Reply: Thank you for your advice. We acknowledge the importance of including a discussion section in our paper. In the revised version, we have incorporated a dedicated discussion section to provide a comprehensive analysis of our findings, explore implications, and discuss the significance of our research in the broader context. This addition aims to enhance the overall depth and completeness of our paper. We appreciate your valuable input and strive to improve the quality of our work. If you have any further recommendations or inquiries, please feel free to inform us. We appreciate your valuable advice once again and will strive to improve the paper. (Line 459-486)

17. Comment: What is the model for 26 surveys?

17. Reply: Thank you for your advice. In our study, we employed the Likert five-point scale method to design the questionnaire. The structuring of these questions was based on the analysis results from the preceding text and aligned with relevant literature. The questionnaire, with a Cronbach’s coefficient of 0.958, surpassing 0.8, demonstrates outstanding reliability in the data. We appreciate your valuable advice once again and will strive to improve the paper.

19. Comment: Extend conclusion, limitation, direction of future research, academic, practical and policy contributions.

19. Reply: Thank you for your valuable suggestions. We acknowledge the need to expand sections such as conclusion, limitations, and future research directions in our paper. In the revised version, we have provided a more comprehensive conclusion, synthesized key findings and emphasizing their implications for both academia and practical applications. Additionally, we have elaborated on the limitations of our study, providing guidance for future research directions. The revised manuscript highlights the academic contributions of our work and its significance in practical applications, offering insights for decision-makers. We appreciate your valuable advice once again and will strive to improve the paper.

Reviewer # 2:

1. Comment: Dear authors, the research proposal is interesting, but I think the authors could improve the text and structure of the article before we consider it for publication. I would like

---

## [Decision Letter · Decision Letter 1]

8 Apr 2024

PONE-D-23-35817R1Mining LDA topics for graded evidence on construction engineering change risks: a data analysisPLOS ONE

Dear Dr. Jin,

Thank you for submitting your manuscript to PLOS ONE. After careful consideration, we feel that it has merit but does not fully meet PLOS ONE’s publication criteria as it currently stands. Therefore, we invite you to submit a revised version of the manuscript that addresses the points raised during the review process.

We look forward to receiving your revised manuscript.

Kind regards,

Saliha Karadayi-Usta, PhD

Academic Editor

PLOS ONE

Reviewers' comments:

Reviewer's Responses to Questions

**Comments to the Author**

1. If the authors have adequately addressed your comments raised in a previous round of review and you feel that this manuscript is now acceptable for publication, you may indicate that here to bypass the “Comments to the Author” section, enter your conflict of interest statement in the “Confidential to Editor” section, and submit your "Accept" recommendation.

Reviewer #1: (No Response)

Reviewer #2: All comments have been addressed

Reviewer #3: All comments have been addressed

2. Is the manuscript technically sound, and do the data support the conclusions?

Reviewer #1: Partly

Reviewer #2: Yes

Reviewer #3: Yes

3. Has the statistical analysis been performed appropriately and rigorously? 

Reviewer #1: No

Reviewer #2: N/A

Reviewer #3: Yes

4. Have the authors made all data underlying the findings in their manuscript fully available?

Reviewer #1: No

Reviewer #2: Yes

Reviewer #3: Yes

5. Is the manuscript presented in an intelligible fashion and written in standard English?

Reviewer #1: No

Reviewer #2: Yes

Reviewer #3: Yes

6. Review Comments to the Author

**Reviewer #1:** The font size and format is strange.

a data analysis in title looks strange.

2.1 Grading of evidence, the title needs to strange.

What is The international GRADE Working Group?

Full name of DIIS process should be used.

State the research gaps reviewing the previous publications: Economic development and construction safety research: A bibliometrics approach, F Luo, Safety science, 2022

Avoid abbreviation like EC risk? What is that?

Include more studies that use text mining approach, Construction safety knowledge sharing on Twitter: A social network analysis Q Yao, Safety science, 2021

Discussion should be based on past literatures.

**Reviewer #2:** The authors extensively revised the original text in line with the guidelines and suggestions of the reviewers. When I analysed the changes made, I could see that the improvements had significantly enhanced the article. In this way, I believe that the text is, with the best judgement, in a position to be evaluated for publication.

Best regards

**Reviewer #3:** In the revised version, the authors have considered the reviewers' comments and it is currently acceptable for publication in the journal.

7. PLOS authors have the option to publish the peer review history of their article (what does this mean?). If published, this will include your full peer review and any attached files.

Reviewer #1: No

Reviewer #2: **Yes: **Dr. Marcio Pereira Basilio

Reviewer #3: No

---

## [Author Response · Author response to Decision Letter 1]

10 Apr 2024

Responses to reviewers 

(Original comments by reviewers are in blue color)

Reviewer #1: 

1. Comment: The font size and format is strange.

1. Reply: Thank you for your feedback on the formatting of our manuscript. We have carefully revised the document according to the journal's formatting guidelines, including adjustments to the font size and style. We appreciate your guidance and hope the changes meet the journal's standards. 

2. Comment: a data analysis in title looks strange.

2. Reply: Thank you for your feedback on our title. We've revised it to "Mining LDA topics on construction engineering change risks based on graded evidence" to better reflect our study's focus and methodology. We appreciate your guidance and are open to further suggestions.

3. Comment: 2.1 Grading of evidence, the title needs to strange.

3. Reply: Thank you for your feedback regarding the title of section 2.1, "Grading of Evidence." It appears there was a misunderstanding or typographical error in your comment, as the term "needs to strange" is unclear. We interpreted this as a suggestion that the title may require adjustment for clarity or appropriateness. To address this concern, we have revised the section title to more accurately reflect the content and purpose of the discussion. The updated title is "Evidence Grading Theory," which directly aligns with the theoretical framework and methodologies discussed in this section. We hope this modification adequately addresses your comment and improves the manuscript's clarity.

4. Comment: What is The international GRADE Working Group?Full name of DIIS process should be used.

4. Reply: Thank you for your queries. The International GRADE Working Group stands for "Grading of Recommendations, Assessment, Development, and Evaluation." It is a collection of methodologists and health care professionals that have developed a systematic approach to grading the strength of evidence and the quality of recommendations in health care. This framework is widely recognized and adopted by health organizations worldwide for its clear, transparent methodology that aids in making informed health decisions.

Regarding the DIIS process, it stands for the sequence of Collecting Data (Data), Revealing Information (Information), Synthesizing Assessment (Intelligence), and Formulating a Solution (Solution). This methodological framework is utilized in our study to ensure a comprehensive and structured approach to decision-making, supporting the effective integration of evidence into the decision-making process.

We appreciate the opportunity to clarify these points and have made the necessary adjustments in the manuscript to reflect this information accurately.

5. Comment: State the research gaps reviewing the previous publications: Economic development and construction safety research: A bibliometrics approach, F Luo, Safety science, 2022

5. Reply: Thank you for your comment. We recognize the importance of F Luo's 2022 analysis on economic development and construction safety using a bibliometric approach, which we have duly cited in our study. Luo's research has provided us with a foundational understanding. Building upon this, our study aims to further investigate the risks of engineering changes in construction engineering, offering updated insights and a more detailed understanding.

We appreciate the opportunity to contribute to this important discussion and hope our research adds valuable perspectives to the field.

6. Comment: Avoid abbreviation like EC risk? What is that?

6. Reply: Thank you for your feedback regarding the use of the abbreviation "EC risk" in our manuscript. To adhere to the journal's guidelines on abbreviations, we introduced "EC risk" as a shorthand for "Engineering Change Risk," given its frequent occurrence in the text. However, recognizing your concern for clarity and the potential for confusion among readers unfamiliar with this term, we have revised the manuscript to either define "EC risk" more clearly at its first . We hope this addresses your concern and improves the readability of our study. 

7.Comment: Include more studies that use text mining approach, Construction safety knowledge sharing on Twitter: A social network analysis Q Yao, Safety science, 2021

7. Reply: Thank you for your suggestion to include more studies utilizing text mining methods. We have incorporated the study by Q Yao (2021) to enrich our research on text mining. This study demonstrates the application of text mining in safety management. We hope these additions further validate the effectiveness of text mining techniques in analyzing unstructured data. We appreciate your guidance and are open to further suggestions.

8.Comment: Discussion should be based on past literatures.

8. Reply: Thank you for your constructive feedback. We acknowledge the importance of grounding our discussion in a comprehensive review of existing literature to ensure the credibility and relevance of our findings. In response to your comment, we have thoroughly revisited the literature and enriched our discussion by incorporating more references to past studies. This has allowed us to better contextualize our results within the broader field and highlight how our work builds upon and contributes to existing knowledge. We believe these enhancements have strengthened the discussion section and thank you for pointing out this opportunity for improvement.

Reviewer #2: The authors extensively revised the original text in line with the guidelines and suggestions of the reviewers. When I analysed the changes made, I could see that the improvements had significantly enhanced the article. In this way, I believe that the text is, with the best judgement, in a position to be evaluated for publication.

Best regards

 Reply: 

We deeply appreciate your thorough review and positive feedback on the revisions we made to our manuscript. It is gratifying to know that the extensive modifications, guided by the insightful recommendations and guidelines provided by you and the other reviewers, have significantly improved the quality of our article. We are encouraged by your assessment and belief that our manuscript is now in a position to be considered for publication.

We are committed to further refining our work as needed to meet the journal's standards and to contribute valuable insights to the field. Your constructive comments have been instrumental in reaching this stage, and we are grateful for the time and effort you invested in reviewing our work.

Thank you once again for your support and encouragement. We look forward to the possibility of our research making a meaningful contribution to the literature.

Best regards

Reviewer #3: In the revised version, the authors have considered the reviewers' comments and it is currently acceptable for publication in the journal.

Reply:

We sincerely thank you for finding the revised manuscript acceptable for publication in the journal. We greatly value the opinions of the reviewers and have made concerted efforts to modify the manuscript in accordance with your suggestions. Being recognized by you is incredibly encouraging.

We look forward to the successful publication of our research and hope it will bring value to the readers and researchers in the field. Thank you for your time, contribution, and for the affirmation of our work

Best regards

---

## [Decision Letter · Decision Letter 2]

25 Apr 2024

Mining LDA topics on construction engineering change risks based on graded evidence

PONE-D-23-35817R2

Dear Dr. Jin,

We’re pleased to inform you that your manuscript has been judged scientifically suitable for publication and will be formally accepted for publication once it meets all outstanding technical requirements.

Kind regards,

Saliha Karadayi-Usta, PhD

Academic Editor

PLOS ONE

Additional Editor Comments (optional):

Reviewers' comments:

Reviewer's Responses to Questions

**Comments to the Author**

1. If the authors have adequately addressed your comments raised in a previous round of review and you feel that this manuscript is now acceptable for publication, you may indicate that here to bypass the “Comments to the Author” section, enter your conflict of interest statement in the “Confidential to Editor” section, and submit your "Accept" recommendation.

Reviewer #1: All comments have been addressed

Reviewer #2: All comments have been addressed

2. Is the manuscript technically sound, and do the data support the conclusions?

Reviewer #1: Yes

Reviewer #2: Yes

3. Has the statistical analysis been performed appropriately and rigorously? 

Reviewer #1: Yes

Reviewer #2: N/A

4. Have the authors made all data underlying the findings in their manuscript fully available?

Reviewer #1: Yes

Reviewer #2: Yes

5. Is the manuscript presented in an intelligible fashion and written in standard English?

Reviewer #1: Yes

Reviewer #2: Yes

6. Review Comments to the Author

Reviewer #1: Abbreviation like ECs should be in full name. More description is needed for EC risk assessment results. How do the Table come about? The Table should be able to read independently. The circles of pyLDAvis results need more details, it is unclear how the diagrams are interpreted.

Reviewer #2: Comments:

This is the third revision I have assessed of this paper; in the last revision the authors had already carried out an extensive revision. At the moment, the authors have improved the text based on the observations of another reviewer. After re-reading the text, I believe that it fulfils the criteria for acceptance. I am therefore satisfied with the changes made by the authors. I therefore reaffirm my last assessment and maintain my recommendation for acceptance in the present form.

Best regards

Reviewer

7. PLOS authors have the option to publish the peer review history of their article (what does this mean?). If published, this will include your full peer review and any attached files.

Reviewer #1: No

Reviewer #2: **Yes: **Dr. Marcio Pereira Basilio

---

## [Editor Report · Acceptance letter]

13 May 2024

PONE-D-23-35817R2 

PLOS ONE

Dear Dr. Jin, 

I'm pleased to inform you that your manuscript has been deemed suitable for publication in PLOS ONE. Congratulations! Your manuscript is now being handed over to our production team.

Kind regards, 

on behalf of

Assoc. Prof. Dr. Saliha Karadayi-Usta 

Academic Editor

PLOS ONE